# Green Extraction of Cellulose Nanocrystals of Polymorph II from *Cynara scolymus* L.: Challenge for a *"Zero Waste"* Economy

Marianna Potenza [1], Laura Bergamonti [1,*], Pier Paolo Lottici [2], Lara Righi [1], Laura Lazzarini [3] and Claudia Graiff [1,*]

1 Department of Chemistry, Life Science and Environmental Sustainability, University of Parma, Parco Area delle Scienze 17/A, 43124 Parma, Italy; marianna.potenza@unipr.it (M.P.); lara.righi@unipr.it (L.R.)
2 Department of Mathematical, Physical and Computer Science, University of Parma, Parco Area delle Scienze 7/A, 43124 Parma, Italy; pierpaolo.lottici@unipr.it
3 Istituto dei Materiali per l'Elettronica ed il Magnetismo, IMEM-CNR, Parco Area delle Scienze 37/A, 43124 Parma, Italy; laura.lazzarini@imem.cnr.it
* Correspondence: laura.bergamonti@unipr.it (L.B.); claudia.graiff@unipr.it (C.G.)

**Abstract:** The increase of agri-food wastes by agriculture and industries is one of the main causes of environmental pollution. Here we propose the recycling of *Cynara scolymus* L. wastes to obtain polymorph II cellulose nanocrystals (CNC). Two different extraction procedures are compared: (i) Soxhlet extraction in an ethanol/toluene mixture, and (ii) water boiling of the agricultural waste. Both procedures were followed by purification of cellulose fibers through bleaching treatments and extraction of cellulose nanocrystals by acid hydrolysis. CNCs have been extensively characterized by FTIR spectroscopy, electrophoretic light scattering measurements, X-ray powder diffraction methods, transmission electron microscopy, and thermogravimetric analyses. Extracted CNC are rod-like-shaped polymorph IIs with a good crystallinity index, and they are characterized by high hydrogen bonding intensity. The ELS measurements on samples from both procedures show good results regarding the stability of the CNC II sol ($\zeta < -40 \pm 5$ mV), comparable to that of the CNC polymorph I. Both polymorph II CNCs show better thermal stability, compared to CNC I. The results show that the easy extraction procedure from agricultural *Cynara scolymus* L. waste can be used to produce high-quality cellulose nanocrystals as a green alternative to the commonly used synthetic route.

**Keywords:** agri-food waste recycling; cellulose nanocrystals of polymorph II; CNC extraction procedure optimization; spectroscopic characterization





## 1. Introduction

The increase of solid waste production from agriculture and industry is one of the main causes of environmental pollution. It has been estimated that about 45% of the world's fruit and vegetable production is wasted [1]. For this reason, the use of biodegradable and renewable wastes represents an opportunity to produce a new generation of materials.

Cellulose nanocrystals are one the most promising new bio-nanomaterials, as pointed out in several papers [2–4].

Most of the lignocellulosic biomass of agri-food origin is mainly composed of cellulose, hemicellulose, and lignin. The efficient separation of the constituent components of these biomasses is one of the major obstacles to efficient use of renewable resources.

The objective of this work is to find a rapid and economic process to obtain stable colloid suspension of cellulose nanocrystals from agricultural wastes.

The *Cynara scolymus* L. plant, known as artichoke, belongs to the Asteraceae family and is typical of the Mediterranean region. The artichokes are globally recognized as a salutary food for their nutritional composition, as well as for high antioxidant properties [5]. The inflorescence, named the head, is the only edible part of the plant. The solid waste consists of the stems and external bracts of the flowers (70–80%). This agri-food residue, of

lignocellulosic nature, is mostly used as green manure and livestock feed, or it is burned [6]. Italy is the world's largest producer of artichokes, producing about 500,000 tons/years, followed by Egypt, Spain, and Peru [7]. The artichoke canning industry produces annually about 900,000 tons of wastes, according to the Food and Agriculture Organization of the United Nations [1]. The 2014 "zero-waste" program of the European Commission encourages the recycling and resource reuse of agri-food residues as relevant sustainable strategies for an efficient circular economy [8]. The most effective way to reduce waste disposal in landfills is to recycle it to produce new materials. Here we propose the extraction from artichoke waste of nanocellulose, a very promising functional bio-nanomaterial, as a sustainable strategy for an efficient circular economy.

Cellulose, a biopolymer made of β-1,4-bound glucopyranose units, is characterized by high-order (crystalline) and low-order (amorphous) regions. Due to the intra- and inter-molecular hydrogen bonds between the hydroxyl groups on the glucopyranose units with oxygen atoms on the same or on neighbor chains, four different crystalline structures (I–IV) are reported in the literature [9]. Native cellulose occurs in two polymorphs: the triclinic Iα mainly found in bacteria and the monoclinic Iβ typical of plants. Both polymorphs consist of parallel chains arranged in packed sheets. In the Iα polymorph, the sheets are directly stacked on top of each other, whereas in the Iβ form, the stacked sheets are staggered between alternating layers [10–12]. When cellulose I is treated with sodium hydroxide or liquid ammonia, cellulose II or cellulose III, respectively, are obtained. Cellulose IV can be produced from cellulose III by a high-temperature treatment in glycerol [13,14].

Cellulose II has a higher chemical reactivity than cellulose I and it is industrially the most relevant. It is characterized by a monoclinic crystal structure with antiparallel stacked sheets, and it contains more intramolecular hydrogen bonds than cellulose I [15,16].

The removal of lignin and hemicellulose to obtain cellulose II from a lignocellulose source can be achieved by means of an alkali solution [13,17,18]. The most used methods are the chemical pulping process, called the Kraft or bisulphite process [19,20], and mercerization, which also has the function of removing hemicellulose and impurities leading to a reorganization of inter-fibrillar regions in the lignocellulosic fibers [21,22]. Treatments with alkalis can improve the mechanical properties of cellulose fibers, such as dimensional stability, fibrillation tendency, tensile strength, dyeability, reactivity, luster, and fabric smoothness [23].

Cellulose is characterized by an intrinsic supramolecular structure that can be isolated as a cellulose nanocrystalline (CNC) species [24]. CNCs can be utilized as building blocks for renewable nanomaterials and new functional bio-nanomaterials [25]. They can be obtained from cellulosic natural sources such as plants, bacteria, animals (tunicate), and algae, and from renewable sources and by-products of agricultural and food processing [26–30]. CNC exhibits fascinating properties such as high strength and high modulus, optical transparency, low coefficient of thermal expansion, biocompatibility, biodegradability, and renewability and low toxicity, which make it suitable for various applications [31].

It is well known [32] that CNC I has better mechanical properties; on the other hand, CNC II seems to be more efficient in terms of functionability. Moreover, it shows better thermal stability, which is an essential requirement for a CNC to be used as additive or filler in composite materials [33]. This arises also from the stronger hydrogen bond network in CNC II compared to CNC I. CNC II can be used as it is or chemically functionalized in the formation of membranes for water pollution, in particular for the adsorption of cationic metals [34]. CNC II can also be used as an adsorber for charged particles and organic dyes [35]. Cellulose nanocrystals of polymorph II allow a wide range of applications, including packaging and synthesis of nanocomposites, drug carriers, food thickeners, and biomedical products [36,37].

In the Table 1, a list of the most important synthesis methodologies and characteristics of CNC polymorph II is reported.

**Table 1.** Synthesis methodologies and characteristics of CNC polymorph II.

| Source | Preparation Method | Length (nm) | CI (%) | Yield% | Reference |
|---|---|---|---|---|---|
| Buckeye cellulose | HCl/$H_2SO_4$ hydrolysis | 500 | 67 | 33 | [38] |
| Bacterial cellulose | $SO_3$/Py | - | 69.9 | 34 | [39] |
| Cellobiose | Cellodextrin phosphorylase from the cellulolytic bacterium | 254 | | - | [40] |
| Lignicellulosic materials | NaOH Mercerization | - | 73 | 67 | [41] |
| Commercial Cellulose | NaOH Mercerization | 120 | 43 | - | [34] |
| Oil palm fronds | HCl hydrolysis | >200 | 47 | - | [35] |
| Jute fibers | TEMPO mediated oxidation and mechanical disgregation | 250 | 56 | - | [42] |
| Softwood pulp | Mercerization | 75 | 55 | - | [33] |
| Cellulose pulp | NaOH Mercerization $H_2SO_4$ hydrolysis | 140 | | - | [43] |

To obtain CNC from purified cellulose sources, the acid hydrolysis is the most used process. In this way, the amorphous regions of the cellulose fibrils are removed to extract the crystalline domains. The commonly used acids are phosphoric, sulfuric, hydrochloric acids, and their mixtures [44]. Sulfuric acid gives CNC suspension with good stability in water, thanks to the presence of negative sulphate surface charges [45].

In this work, an approach of a circular economy that is "*zero waste*" by the alternative, green extraction of cellulose nanocrystals of polymorph II from *Cynara scolymus* L. is proposed. The removal process of lignin and hemicellulose is optimized to obtain cellulose pulp in a cheap, simple, and fast way, comparing two different methods. The first procedure (i), inspired on [30] changing some parameters, is based on the use of a Soxhlet apparatus, in the presence of different solvents and reagents; the second procedure (ii) is carried out simply boiling the lignocellulosic materials in an alkaline medium. The latter is a green method: no solvent harmful to the environment is used; no polluting waste is produced. As known, a very efficient whitening is obtained by using NaClO (at a low concentration to avoid strong oxidation of the cellulose). A less aggressive method using a mixture of NaCl and $CH_3COOH$ was preferred. To ensure effective purification of the extracted pulp, after bleaching, a further treatment with sodium bicarbonate was carried out to remove the hemicellulose component [46]. In the procedure (ii), a new, easy, and ecofriendly extraction method is proposed, experimenting on a single bleaching/purification step with sodium carbonate.

The extracted CNCs have been fully characterized: the CNC sols stability was evaluated by ELS measurements; TEM was employed to investigate the morphology and size of nanofibers; the complex intra- and intermolecular hydrogen bonding system and characteristics of cellulose I and II were explored by FTIR spectroscopy; structural details of the cellulose nanofibers (polymorphs identification, preferred orientation and amorphous fraction, and crystallinity index) were obtained by an X-ray powders diffraction investigation; diffraction data were analyzed by profile fitting procedures to determine the crystalline size of the extracted cellulose fibers; thermogravimetric analysis was carried out to assess the thermal stability. The CNCs obtained by both procedures were compared to cotton-based CNC (defined as CNC-CF) and mercerized CNC (CNC-MF).

## 2. Materials and Methods

### 2.1. Materials

*Cynara scolymus* L. was acquired from a local market. Cotton linter pulp was purchased from Parchin Chemical Industries Co., Tehran, Iran. Sodium chloride (99.9%, Merk, Darmstad, Germany), sodium hydroxide (Carlo Erba, Emmendingen, Germany), sodium bicarbonate (Carlo Erba, Emmendingen, Germany), glacial acetic acid (100%, Sigma-Aldrich, Darmstad, Germany), and ethanol (96%, Sigma-Aldrich, Darmstad, Germany) were used for separation of cellulose from Cynara stems and bracts. Sulfuric acid

(95–97%, Merk, Darmstad, Germany), hydrochloric acid (37%, Fluka, Darmstad, Germany), and phosphoric acid (85–90%, Fluka, Darmstad, Germany) were utilized for hydrolysis. Chemical reagents and solvents were used as received without further purification.

### 2.2. Cellulose Isolation Procedures

*Cynara scolymus* L. consists of cellulose (35%), hemicellulose (16%), lignin (17%), ashes (8%), and other compounds (polyphenols, fatty acids, pigments, and various impurities) (24%) [30].

The extraction of cellulose fibers requires pre-treatment steps, including alkaline treatments and bleaching. First, the bracts and stems of Cynara were washed in lukewarm tap water, dried in a ventilated oven at 40 °C for 24 h, and finely chopped. At this stage, two different procedures have been followed to eliminate non-cellulosic components:

Procedure (i): dried, small Cynara pieces were extracted with a solution of toluene/ ethanol (2:1, $v/v$) in a Soxhlet apparatus for 6 h; the solid residue was dried in a vacuum pump, then extracted in ethanol for 2 h to eliminate the residual toluene, and again dried under vacuum. The use of mixtures of polar and non-polar solvents allows to remove both soluble and insoluble fractions (mainly waxes and fatty acids). The solid residue was then delignified by a 2.5 M NaOH solution (using solid to solution ratio of 1/8 ($w/v$) at 95 °C for 2 h). The pulp bleaching was performed by using sodium chloride and acetic acid, in a molar ratio of 4/1. The dried pulp was dispersed in distilled water and heated to 80 °C, then, at regular intervals of 30 min, an aliquote of a bleaching agent (about 5/1 with the pulp) was added. The procedure was repeated 6 times. The resulting pulp was recovered, repeatedly washed, and dried in an oven at 40 °C for 24 h. To ensure effective removal of hemicellulose, the holocellulose pulp was purified by dispersing the residue in a sodium bicarbonate solution (1 M) at 80 °C for 3 h. Finally, the solid residue was separated by centrifugation, washed, filtered, and dried. The final product was called Cel-S.

Procedure (ii): dried, small Cynara pieces were dispersed in distilled water at 80 °C for 2 h. The residue was dried at 40 °C for 24 h. The cycle was repeated two times. The pulp delignification was carried out in a 5 M NaOH solution (using solid to solution ratio of 1/8 ($w/v$) at 80 °C for 2 h) and then washed in distilled water and dried for 24 h at 40 °C. Sodium carbonate, a dry pulp/$Na_2CO_3$ ratio of 1/10 ($w/w$), was used both to leach from hemicelluloses and to bleach solid pulp. Finally, the dispersion was heated to 80 °C and vigorously stirred for about 4 h, until the solid became white. The dispersion was centrifuged, and cellulose fibers were washed and dried. The final product was named Cel-NS.

### 2.3. Cellulose Nanocrystals Preparation

The cellulose pulp samples obtained by the procedures described above were subjected to sulfuric acid (40% $w/v$) hydrolysis to separate the nanocrystalline phase. An acid/cellulose ratio of 8.75 ($w/w$) was chosen. The hydrolysis reaction was carried out at 45 °C for 30 min [47], then quenched with iced water. The obtained suspension was then centrifuged at 6000 rpm for 10 min to eliminate excess water and acid. The precipitate was suspended in distilled water and centrifuged until an opalescent suspension was obtained. The purification of the CNC suspension was carried out using dialysis membrane tubes (cut-off of 10–12 kDa, Sigma-Aldrich) in ultrapure water, changing water every 24 h, up to a neutral pH. The cellulose suspension was sonicated for 30 min using a horn ultrasonicator. The CNCs obtained from Cel-S and Cel-NS were named CNC-S and CNC-NS, respectively. The procedure scheme was graphicated in Figure 1.

For comparison, CNC suspensions were prepared by hydrolysis with sulfuric acid (64%) [48] starting from raw cotton fibers (CNC-CF) or mercerized fibers (CNC-MF). The mercerization was attempted by treatments with an NaOH solution (20%) at 60 °C for 2 h. To remove NaOH, the samples were washed repeatedly until reaching a neutral pH.

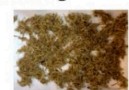

**Stems and bracts preparation**

- Washing (cold water)
- Drying (40 °C, 24 hours)
- Grinding

**Extractives removal**

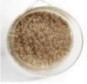

| **Procedure (i)** **Soxhlet** | **Procedure (ii)** **No Soxhlet** |
|---|---|
| • Toluene/Ethanol (1:1, v/v) 6h | • Water, 80 °C, 2h |
| • Drying in vacuum | • Drying in oven 40 °C 24h |
| • Washing in Ethanol 2h | **2 times** |
| • Drying in vacuum | |

**Lignin removal**

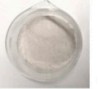

- 2.5 M NaOH solution (solid/solution ratio 1/8 (w/v) 95 °C for 2h)
- Drying 40 °C, 24 h

- 5 M NaOH solution (solid/solution ratio 1/8 (w/v) 80 °C for 2h)
- Drying 40 °C, 24 h

**Bleaching**

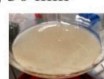

- Pulp dispersion in water, 80 °C
- NaCl/Acetic Acid 4/1, pulp/beaching agent ratio 1/5
- Drying 40 °C, 24 h
- NaHCO$_3$ (1M) 80 °C, 3h

- Pulp dispersion in water, 40 °C
- Sodium carbonate/pulp ratio 10/1 (w/w)
- 80 °C, 4h

**CNC isolation**

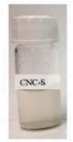

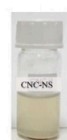

- Acid Hydrolysis (H$_2$SO$_4$ 40%) 45 °C, 30 min

- Centrifugation
- Dialysis

**Figure 1.** Representative scheme of the conventional (i) and innovative (ii) procedure to obtain CNC-S and CNC-NS, respectively.

All the CNC suspensions were cast into glass Petri dishes and dried in an oven at 40 °C for 24 h.

The yield (%) of obtained CNCs was calculated according to the following equation:

$$\text{Yield } (\%) = m_{(f)}V_1 m_{(f)}V_1 / m_{(i)}V_2 \ \times 100 \tag{1}$$

In this equation, $m_f$ is the mass of the vacuum-dried sample (g), $m_i$ is the initial mass of dry cellulose sources (g), $V_1$ is the volume of the total CNCs suspension after dialysis (mL), and $V_2$ is the volume of CNCs suspension which was dried with vacuum (mL).

The obtained yield for CNC-CF, CNC-MF, CNC-S, and CNC-NS was 83%, 32%, 25%, and 26%, respectively [2,49,50].

### 2.4. Methods

### 2.4.1. Electrophoretic Light Scattering (ELS) Measurements

The surface charge (zeta potential) of the CNC particles was determined by electrophoretic light scattering (ELS). A Brookhaven 90 Plus Particles Size analyzer operating

in the particle range of 0.6 nm–6 μm was used. All sols were diluted in distilled water in a ratio of 1:10 ($v/v$), with a final concentration of about 0.20 mg/mL.

### 2.4.2. TEM Analysis

Transmission electron microscopy (TEM) observations, to investigate the morphology and size of nanofibers, were carried out with a JEOL 2200FS field emission microscope operating at 200 kV accelerating voltage, equipped with two High Angle Annular Dark Field detectors (HAADF) and the Energy Dispersive X-ray Spectrometer (EDX). All the investigations were performed in the HAADF Scanning TEM (STEM) mode in order to detect the Z contrast in the images and to obtain compositional maps of the samples by EDX. The samples for the observations were prepared by dropping a diluted (0.01, $w/w\%$ in water) and sonicated suspension of the CNCs onto 300-mesh holey carbon copper grids.

### 2.4.3. X-ray Powder Diffraction Analysis (XRPD)

Powder diffraction measurements were carried out to ascertain the polymorphic variant featuring the nanocrystalline cellulose. Furthermore, the crystal size of the nanostructured phases of cellulose was established with a full profile fitting procedure. The X-ray diffraction characterization was performed with a Thermo ARL X'TRA X-ray diffractometer with Si-Li detector, using Cu-Kα radiation at 40 kV and 40 mA. The samples were scanned in a 2θ range of 5–60°. The crystallinity index (CI%) of the CNCs was calculated according to the method proposed by Segal [51]:

$$CI\% = [(I_{200} - I_{am})/I_{200}] \times 100 \tag{2}$$

where $I_{200}$ is the maximum peak intensity corresponding to the (200) reflection at $2\theta \approx 22°$ for cellulose II and $\approx 23°$ for cellulose Iβ, which represents the crystalline region, while $I_{am}$, which concerns the amorphous portion, is the minimum intensity value at $2\theta \approx 16 \div 18°$ between the peaks relative to the (110) and (200) reflections [52].

In order to gain further structural details of the cellulose nanofibers in terms of crystal size, polymorph identification, preferred orientation, and amorphous fraction, a Rietveld refinement was carried out on the overall X-ray diffraction pattern. To define the diffraction peaks shape, a pseudo-Voigt profile function was adopted where the gaussian contribution ($G_{(\theta)}$) is expressed by:

$$G_{(\theta)} = GUtan^2\theta + GVtan\theta + GW + GP/cos^2\theta \tag{3}$$

GU, GV, and GW correspond to the Caglioti parameters describing the instrumental contribution of the variation in full-width at half maximum (FWHM) as a function of the diffraction angle (obtained by the fitting of the standard $Al_2O_3$). The GP term is used in the Scherrer equation:

$$CS \text{ (crystal size)} = K\lambda/\pi\sqrt{GP} \tag{4}$$

where CS is the average diameter of the crystals, K is a form factor (taken 0.90 in spherical approximation), and λ is the wavelength of the incident beam. GP is the FWHM correlated to the integral width for each reflection and dependent from the CS.

### 2.4.4. FT-IR Spectroscopy

The typical functional groups and the hydrogen bonding system of CNCs were studied by FTIR spectroscopy. The FTIR spectra, in ATR mode, were obtained using the Thermo-Nicolet Nexus spectrometer equipped with Thermo Smart Orbit ATR diamond accessory, in the range 4000–400 cm$^{-1}$, with a spectral resolution of 4 cm$^{-1}$. Curve fitting was carried out by LabSpec 5.78.24 software (Jobin Yvon Horiba, Kyoto, Japan), after background subtraction by a 2nd degree polynomial. The number of peaks involved were determined based on the second derivative of FTIR spectra in the range 3300–3700 cm$^{-1}$ [53].

2.4.5. Thermogravimetric Analysis (TGA)

Thermogravimetric analysis was carried out to assess the thermal stability. TGA analyses were carried out by means of a Perkin Elmer TGA8000 instrument (mass sample: 1–3 mg) at a heating rate of 10 °C·min$^{-1}$ in the temperature range 30–550 °C. The measurements were performed at an atmospheric pressure under air atmosphere.

## 3. Results

### 3.1. ELS Measurements

The stability of the sols was evaluated by ELS measurements (Table 2). All cellulose nanocrystals suspensions are negatively charged because of the sulfate groups on the surface due to the esterification with hydroxyl groups and sulfuric acid. These groups contribute to the stabilization of the suspensions thanks to electrostatic repulsion [45]. Generally, values of the zeta potential outside the −30 mV/+30 mV range indicate good sol stability [37]. The results indicate good stability for all the CNCs extracted from *Cynara scolymus* L. compared to CNC from cotton linter. In particular, ELS measurements suggest high sol stability of CNC-NS, in perfect agreement with that of CNC-CF. Probably, the procedure (i) does not have a high efficacy in terms of lignin removal; therefore, the acid hydrolysis does not bring the same amount of charge as the procedure (ii) due to the greater presence of amorphous regions. The CNC-S Zeta potential is therefore comparable to that of the CNC-MF.

**Table 2.** Zeta potential of CNC suspensions.

|  | CNC-S | CNC-NS | CNC-CF | CNC-MF |
| --- | --- | --- | --- | --- |
| Zeta potential (mV) | −33 | −45 | −51 | −32 |

### 3.2. TEM

The conventional TEM imaging techniques could not be used to obtain information on these samples, due to the poor contrast. All the samples have been investigated by means of the HAADF STEM technique, so-called Z contrast, that allows to enhance the contrast in the images between regions with different average atomic numbers and/or different amounts of material. Figure 2 represents the comparison between the samples CNC-CF, CNC-NS, and CNC-S. CNC-S and CNC-CF are similar. The crystals appear in the form of rod-like whiskers, as reported in the literature [54]. Their dimensions are comparable: the average length is about 200 nm, while they are 20 nm or less thick. In comparison, CNC-NS sample presents tangled crystals, possibly embedded in amorphous material, that could not be studied individually. The images of CNC-S and CNC- NS appear noisy due to the need of using a short acquisition time to avoid the fast contamination of the samples under the beam irradiation.

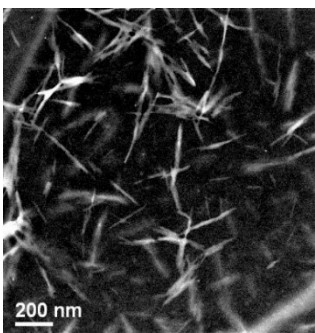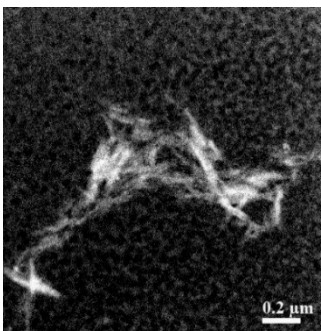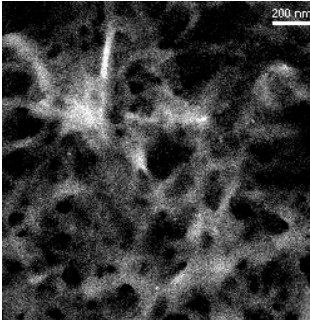

**Figure 2.** HAADF-STEM images of the samples of CNC-CF, CNC-S, and CNC-NS, from left to right.

*3.3. XRPD Analysis*

The cellulose polymorph Iβ can be irreversibly converted into the form II by mercerization with an NaOH aqueous solution followed by washing and recrystallization. In this work CNC-S and CNC-NS were obtained after NaOH treatment of *Cynara Scolymus* L. waste and CNC-MF after mercerization of pure cotton fiber cellulose. Accordingly, the nanocrystals show the cellulose II phase. On the other hand, CNC-CF, obtained by acid hydrolysis of pure cotton, exhibits a cellulose Iβ phase. Although both Iβ and II polymorphs possess the same monoclinic symmetry with the $P2_1$ space group, they are characterized by a different spatial distribution of the polymeric cellulose chains [11,15].

In the two crystal structures, the chains are aligned along the *c* axis, but, as depicted in Figure 3, they are characterized by a different packing mediated by intermolecular hydrogen bonds. The two crystal structures and corresponding atomic positions were obtained from earlier literature [11,12].

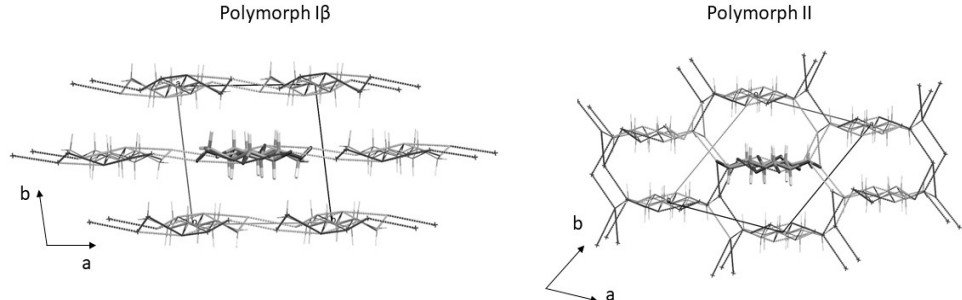

**Figure 3.** Projection along the *c* axis for both polymorphic forms observed in the CNC analyzed.

In the Iβ polymorph, the planar sheets of polysaccharide chains are stacked along the *b* axis, whereas the phase II is constituted of a honeycomb-like distribution of the chains. The detailed structure may be found in literature [11,55]. The unit cell parameters for the two polymorphic phases calculated for the samples here analyzed are listed in Table 3.

**Table 3.** Unit cell parameters obtained for the two crystalline phases Iβ and II featuring the investigated CNC suspensions.

|  | Iβ Phase CNC-CF | II Phase CNC-MF | II Phase CNC-S | II Phase CNC-NS |
|---|---|---|---|---|
| Symmetry | $P2_1$ | $P2_1$ | $P2_1$ | $P2_1$ |
| *a* (Å) | 7.826 (6) | 8.033 (4) | 8.094 (4) | 8.090 (3) |
| *b* (Å) | 8.316 (5) | 9.056 (5) | 9.041 (5) | 9.069 (4) |
| *c* (Å) | 10.349 (7) | 10.157 (6) | 10.171 (7) | 10.37 * |
| $\gamma$ (°) | 95.10 (6) | 118.14 (8) | 118.14 (9) | 118.17 (2) |

* the *c* parameter was not refined, owing to the limited scattering from the polymorph II.

The first two samples analyzed by XRPD were cotton-based CNC-CF and CNC-MF, used as standard of forms Iβ and II, respectively, to define a structural analysis protocol. Figure 4 displays the Rietveld refinement based on the crystal structure for the Iβ phase [11] and featured by the three intense peaks at 2θ–15°, 16.7°, and 23°.

The fitted background, obtained by a polynomial function, is composed by a prominent bump at 2θ–22° which corresponds to the contribution of the amorphous fraction of cellulose [56].

The Rietveld refinement converges to a value of GP, indicating a crystal size of approximately 16 nm for the Iβ nanocrystalline cellulose. It is worthy of notice that, in the case of elongated nanocrystals, it is only possible, with classical XRPD experiments, to estimate the short size of the nanoparticles. The (200) reflection is featured by a prominent intensity related to the preferred orientation of the nanofibers. By introducing March–Dollase correction for preferred orientation, an evident improvement of the structural refinement yields

final agreement factors Rp = 3.5% and Rwp = 4.8%. Such an effect is expected, considering that the cellulose fiber grows along the *c* axis and the reflections showing appreciable intensity belong to the hk0 class.

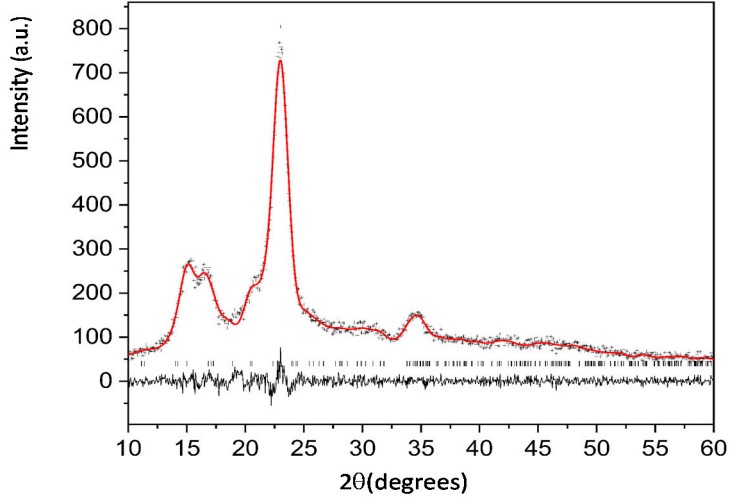

**Figure 4.** Rietveld analysis for the phase Iβ characterizing the CNC-CF. The ticks mark the reflections of the monoclinic crystal structure, and the red curve indicates the calculated fitting.

The Rietveld refinement for the polymorph II of the cotton-based CNC standard was performed following the same procedure. The CS approaches 10 nm, indicating a slight decrease of the crystallites dimension if compared to the Iβ counterpart. Seemingly, the mercerization process required for the crystallization of the II form determines unfavorable conditions for the growth of the crystalline domains.

The XRPD patterns collected on the CNC samples displayed in Figure 5 indicate that the cellulose shows the form II as a single phase. The CNC-NS is constituted by a large fraction of amorphous cellulose with a minute crystal size (CS) for the nanocrystalline phase. The lack of an appreciable fraction of nanocrystals prevented the accurate structural analysis and, in this case, we performed a Le Bail profile fitting to determine only unit cell constants and the CS parameter as well. Hence, the polymorph II is therefore composed by minute crystals ranging from 5 to 9 nm.

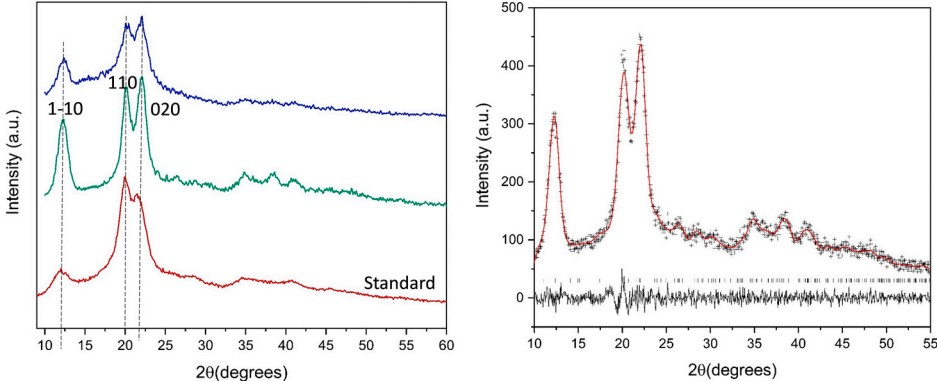

**Figure 5.** (**Left**): XRD for the mercerized CNC samples showing the typical pattern of the polymorph II (red-CNC-MF standard, green-CNC-S, Blue-CNC-NS). (**Right**): Rietveld plot for the structural refinement of CNC-S.

Interestingly, the Rietveld analysis of the XRPD pattern (see Figure 5) for CNC-S indicates the presence of the phase II with an average dimension of nanocrystals of 13 nm. As already stressed for the Iβ form, the determination of the domain extension of the

mercerized crystals is limited to the ab plane perpendicular to the fiber's growth direction. Nevertheless, the crystals do not exhibit a preferred orientation and no correction was introduced during the Rietveld refinement.

Crystallinity index (CI%) was determined using the conventional Segal method (Equation (2)) [51]. The CI% for CNC-S and CNC-NS is 75% and 63%, respectively, in agreement with [57,58]. Although the crystallinity index found for the CNC-NS sample is the lowest, these results are in agreement with those found for the standard of polymorph II, in which CNC-MF has a CI% (67%) suggesting that both methods studied gave good extraction results. The CI% obtained for CNC-CF (89%) is higher because of the use of a pure cotton, raw cellulose material and a fewer number of extraction steps according to literature [47,58]. The CI% results are summarized in Table 4.

**Table 4.** Crystallinity index (CI%) of CNCs calculated with the Segal method.

| Sample | Crystallinity Index (CI%) |
|---|---|
| CNC-NS | 63 |
| CNC-MF | 67 |
| CNC-S | 75 |
| CNC-CF | 89 |

### 3.4. FTIR Spectroscopy

FTIR spectra of CNC samples, isolated from raw and mercerized cotton linter and from *Cynara scolymus* L., are shown in Figure 6. All spectra were normalized with respect to the band associated with the CH stretching vibrations in the 2750–2980 cm$^{-1}$ range. All CNCs are dominated by the characteristic bands of the cellulose associated with alcohol groups vibrations: a strong-broad band due to $\nu$O-H in the 3500–3200 cm$^{-1}$ region and very strong bands due to $\nu$C-O in the 1200–900 cm$^{-1}$ region. As clearly observable in the CNC-S and CNC-NS spectra, both the cellulose extraction procedures from *Cynara scolymus* L. wastes were effective. The features of lignin components, the $\nu$C-O of acetyl and ester groups at 1730 cm$^{-1}$, the $\nu$C = C of the aromatic ring at about 1600–1510 cm$^{-1}$, and the $\nu$C-O of aryl the groups at 1240 cm$^{-1}$ [59,60] are absent.

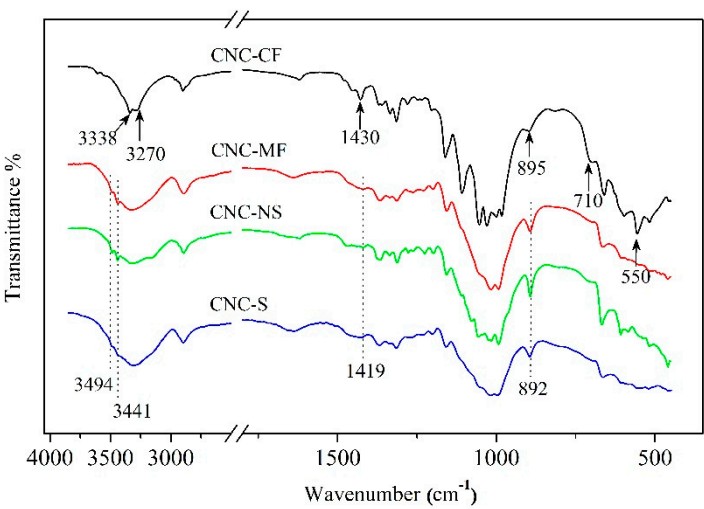

**Figure 6.** FTIR spectra of CNC isolated from cotton linter (CNC-CF and CNC-MF) and from *Cynara scolymus* L (CNC-NS and CNC-S).

Due to the alkali treatments of cellulose pulp from cotton (CNC-MF) and from Cynara (CNC-S and CNC-NS), the cellulose polymorph II is obtained. The FTIR spectra show the characteristic peaks at 3494 cm$^{-1}$ and 3441 cm$^{-1}$ due to the stretching vibrations of OH groups engaged in H-bonds, the band at 1419 cm$^{-1}$ due to CH$_2$ bending and the

band at 892 cm$^{-1}$, and the characteristic of the polymorph II, assigned to νC-O-C of the β-glycosidic linkage [15]. The two last bands are found in the polymorph Iβ at 1430 cm$^{-1}$ and 895 cm$^{-1}$, respectively [16].The CNC-CF FTIR spectrum, reported for comparison, exhibits the features of the Iβ structure, with the identifying bands at 3338 cm$^{-1}$ and 3274 cm$^{-1}$ associated to intra-molecular hydrogen bonds, and the features at 1430 cm$^{-1}$ and 710 cm$^{-1}$ assigned to CH$_2$ bending and rocking vibrations, respectively [47], as well as the band at 550 cm$^{-1}$ attributed to C6-OH torsion. The main changes between the polymorph I and II, due to the different inter- and intramolecular bonds, are in the νOH vibration region and in the 1500–600 cm$^{-1}$ fingerprint region, where band shifts and absorbance intensity variation can be noted. In Table 5 are reported the main FTIR bands and their assignments [15,16,61].

**Table 5.** The assignment of the main FTIR bands of CNC Iβ (CNC-CF) and CNC II (CNC-MF, CNC-S, CNC-NS) according to references, and the band shift and absorbance intensity variation from cellulose Iβ and cellulose II.

| Wavenumber (cm$^{-1}$) | | Δν (cm$^{-1}$)/Absorbance Change | Assignment |
|---|---|---|---|
| **CNC Iβ** | **CNC II** | | |
| 3338 | 3494 | +156/- | νO3H—O5 intramolecular H bonds |
| 3270 | 3441 | +171/- | νO2H—O6 intramolecular H bonds |
| 2900 | 2887 | −13/- | νCH |
| 2850 | | | νCH |
| 1482 | 1470 | −12/Δ | δCH$_2$ |
| 1430 | 1419 | −10/∇ | δCH (C1) |
| 1366 | 1373 | +7/∇ | δC–H |
| 1336 | 1336 | -/∇ | δCOH in plane (C2 or C3) |
| 1236 | 1226 | −10/Δ | δCOH in plane (C6) |
| 1203 | 1195 | −8/- | δCOH in plne |
| 1160 | 1153 | −7/∇ | ν$_{as}$COC (β-glycosidic bond) |
| 1030 | 1014 | −16/∇ | νC-OH of C6 |
| 984 | 994 | +10/Δ | C-O valence vibration of C6 |
| 895 | 892 | −3/Δ | COC of the ν-glycosidic bond and/or ether group |
| 660 | 665 | −5/- | δCOH out of plane |
| 550 | | | τOH of C6 |

Key to symbols: ν: stretching, δ: bending, τ: torsion, Δ: increase, ∇: decrease, -: equal.

Cellulose II is characterized by a complex intra and inter-molecular hydrogen bonding system that forms a tightly compact 3D structure. An empirical relationship known as Hydrogen Bond Intensity (HBI), proposed by Nada et al. [62], allows obtaining information on the degree of inter- and intramolecular H-bonding. This relationship is based on the ratio of the absorption bands at 3336 cm$^{-1}$ and 1336 cm$^{-1}$ (associated with the crystalline order). In agreement with several authors [16,63], it is observed that the CNC-CF is characterized by higher crystallinity and a lower HBI (1.47(3)), compared to the mercerized CNCs (CNC-MF = 2.39(2); CNC-S = 1.98(2), CNC-NS = 2.18(4). As observed in Goswami et al. [63], there is an inverse relationship between CI and HBI, according to results reported in Table 3, where the higher CI% correspond to CNC-CF (89%). The alkali treatment during the mercerization process, as suggested by Široký et al. [16], causes the breaking of the intramolecular H-bond that stabilizes the β-glycosidic linkage in crystalline cellulose. Because of the conversion of the cellulose Iβ to cellulose II, voids and pores within the fibers and crystalline structure are modified.

Patterns of hydrogen bonding network in the cellulose polymorphs Iβ and II are reported in Figure 7. Each glucopyranose unit has three free hydroxyl groups that can be involved in different inter- and intra-molecular hydrogen bonds with neighboring molecules (of the same or adjacent chain). Hydroxyl groups can act both as proton donor and as proton acceptor. Inter-molecular hydrogen bonds are nearly orthogonal to the axis of the cellulose chains, while intra-molecular H-bond are almost parallel.

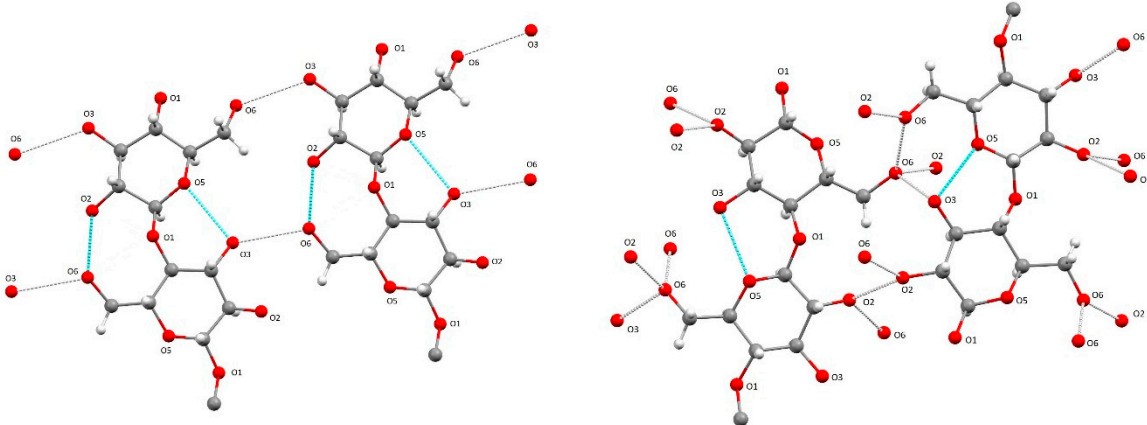

**Figure 7.** Patterns of inter- and intramolecular bonds in cellulose Iβ (**left**) and II (**right**). The cyan and light green dotted lines indicate intra- and intermolecular hydrogen bonds, respectively.

In Iβ cellulose, intermolecular hydrogen bonds mainly engage primary alcohols as proton donors, whereas intramolecular hydrogen bonds are preferably established by secondary alcohols. In the polymorph II, the hydroxyl groups are involved in a complex 3D network of H-bonding between the antiparallel and staggered chains. In the vibrational spectra, the strong and broad-band associated to the OH stretching vibration is the result of the convolution of several sub-bands, as shown in Figure 8 [64,65].

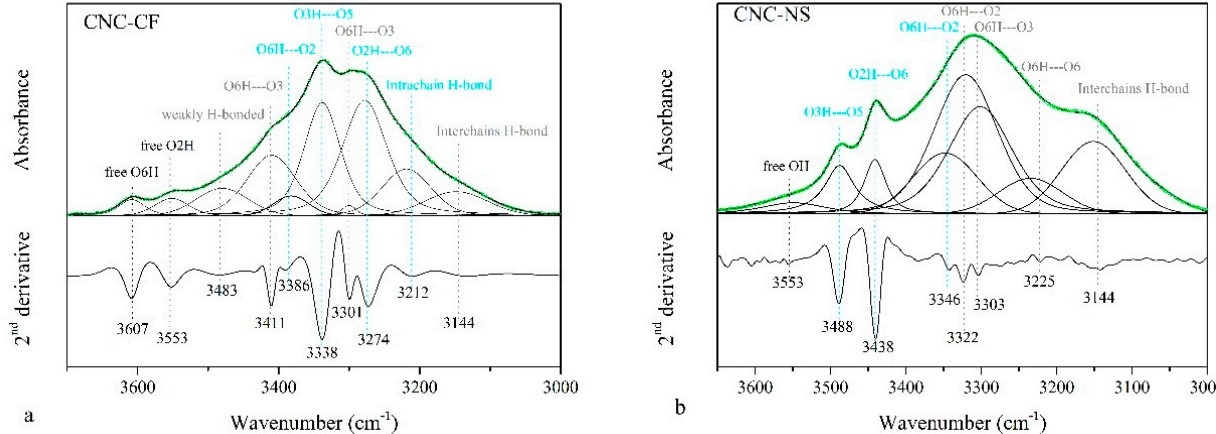

**Figure 8.** Deconvolution of the FTIR spectra and second derivative of (**a**) CNC-CF (polymorph Iβ) and (**b**) CNC-NS (polymorph II) in the 3700–3000 cm$^{-1}$ region. The cyan and light gray colors refer to intra- and intermolecular H-bonds, respectively.

### 3.5. Thermal Analysis

Thermal analysis was carried out to assess the thermal stability and degradation profiles of the CNC samples. The TG and DTG curves of the CNCs in Figure 9a,b show that the decomposition of the CNC fibers occurs at different temperatures, denoting the presence of distinct components. All TGA curves show a small initial deflection, between 60 °C and 150 °C. The initial weight loss (less than 10%) is similar in all CNCs and can be mainly attributed to the loss of moisture absorbed by the surface, including chemisorbed water and/or hydrogen-bound water [66]. In the high-temperature range (>150 °C), the thermal degradation behavior of CNC I and CNC II is quite different. For the first sample, two pyrolysis processes are observed in the DTG curve between 180–300 °C (40% weight loss) and 300–500 °C (20% weight loss), attributed to sulphate groups on the crystal surface and to the degradation of the network cellulose, respectively [67]. The presence of sulfate ester groups, replacing the hydroxyl groups on the surface of the nanocrystals, provide a negative

charge to CNCs which stabilizes the aqueous suspension against flocculation, but they also compromise the thermal stability of nanocrystals, as also observed by Naduparambath et al. [68], and as reported by Chieng et al. [69]. This leads to a decrease in the activation energy due to CNC degradation, making the sample more sensitive to thermal degradation. The greater quantity of the sulphate groups is confirmed by the ELS analysis (−51 mV). Furthermore, as observed in the TEM images, the smaller nanocrystal dimensions of the CNC-CF compared to other CNCs provide a high surface area of the nanocrystals, which could play an important role in decreasing their thermal stability [68]. In the case of cellulose II nanocrystals (CNC-S, CNC-NS, and CNC-MF) the degradation process is shifted to higher temperatures (270–400 °C), probably due to the stronger interaction of the -OH groups, which require more energy to initiate the degradation process. For all the samples, there was a weight loss ranging from 56% to 68%. The hydroxide ions penetrate widely inside the crystals during the alkaline treatment, producing a swelling of the CNC [32]. After the removal of the ions during dialysis, the cellulose chains recrystallize into cellulose II, which is thermodynamically more stable. Furthermore, for mercerized CNCs, as also observed by Johar [70], the removal of hemicellulose and lignin improves the thermal stability of the materials. The latest weight loss above 500 °C is due to the rapid depolymerization of carbon residues.

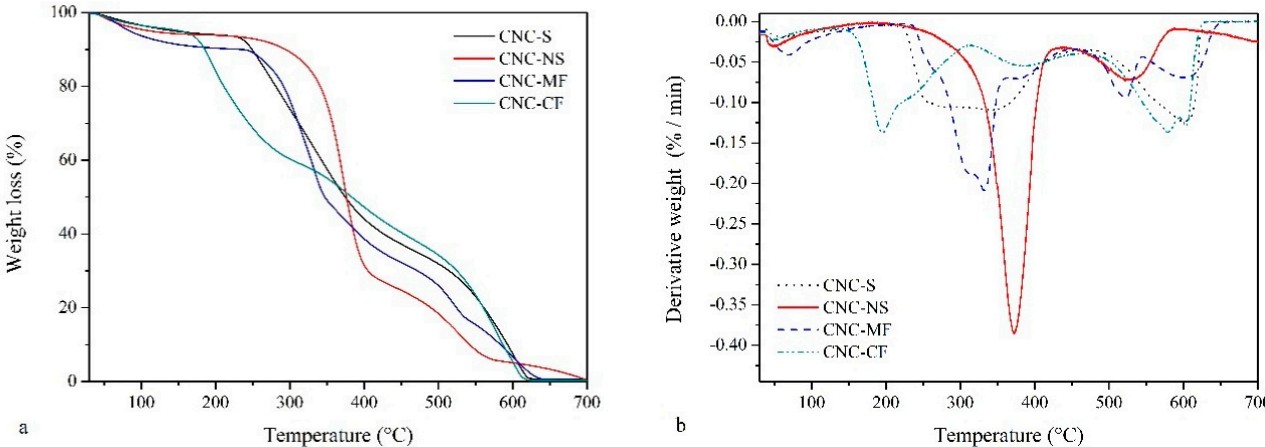

**Figure 9.** TGA (**a**) and DTG (**b**) curves for CNC's powders.

## 4. Conclusions

A novel eco-friendly extraction of cellulose from *Cynara scolymus* L. waste has been successfully carried out to obtain mercerized cellulose nanocrystals of polymorph II.

The easy procedure developed allows the obtaining of the raw material purification by simple boiling using water as a solvent as an alternative to an ethanol/toluene mixture of the typical Soxhlet method. With both tested procedures it is possible to obtain a suspension of high-quality cellulose nanocrystals after delignification in an alkaline medium, followed by acid hydrolysis, centrifugation, and dialysis. All CNCs show rod-like-shaped crystals about 200 nm long and about 20 nm thick. CNC-S and CNC-CF are comparable, while the CNC-NS sample features tangled crystals embedded in amorphous material, like the lower crystallinity index (63%) suggests. All sols have good stability in water, but the new extraction method gives the best results (ζ −45 mV) comparable to those of CNC-CF. The cellulose II nanocrystals (CNC-S, CNC-NS, and CNC-MF) show high thermal stability; the degradation process is shifted to higher temperatures than CNC I.

With a view to a zero-waste circular economy, the extraction of cellulose from *Cynara scolymus* waste proposed here can be an effective alternative to recycling waste from the agri-food industry. Furthermore, the suggested method of cellulose extraction, to obtain CNC II, could be developed to an industrial level, avoiding the use of hazardous and environmentally harmful solvents.

**Author Contributions:** Conceptualization, C.G. and L.B.; methodology, L.B., M.P., L.R. and L.L.; validation, L.B. and P.P.L.; investigation, M.P.; data curation, P.P.L.; writing—Original draft preparation, L.B., L.R., L.L. and M.P.; writing—Review and editing, P.P.L. and C.G.; supervision, C.G. and P.P.L. All authors have read and agreed to the published version of the manuscript.

**Funding:** This research received no external funding.

**Institutional Review Board Statement:** Not applicable.

**Informed Consent Statement:** Not applicable.

**Data Availability Statement:** Not applicable.

**Acknowledgments:** Giovanni Predieri is gratefully acknowledged for useful discussion and Francesca Pintabona for experimental work during her degree thesis. Cartiere di Guarcino S.P.A. is gratefully acknowledged for its collaboration. This work has benefited from the equipment and framework of the COMP-HUB Initiative, funded by the 'Departments of Excellence' program of the program of the Ministero dell'Istruzione, della Università e della Ricerca Italian Ministry for Education, University and Research (MIIUR-Italy, 2018–2022).

**Conflicts of Interest:** The authors declare that they have no known competing financial interests or personal relationships that could have appeared to influence the work reported in this paper.

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
