# Peer review of "Green Extraction of Cellulose Nanocrystals of Polymorph II from Cynara scolymus L.: Challenge for a “Zero Waste” Economy"

_crystals, doi:10.3390/cryst12050672_

Round 1

Reviewer 1 Report

The manuscript provides good insight toward the production of CNC from Cynara scolymus L and studied the crystallinity using XRD and TEM and provided charesteristic functional groups presence through FTIR. Overall the manuscript can be accepted for publication provided following queries raised below:

  1. Regarding the crystallinity index (CI%), as authors suggested CNC-S has better crystallinity compared to CNC-NS, why there is no attempt to improvise the crystallinity of CNC-NS with other known methodology?
  2. We would be happy to see Calculations of CNCs yield and residue yield for all four type.
  3. As author claims, “an approach of circular economy “zero waste” by alternative green extraction of cellulose nanocrystals polymorph II from Cynara scolymus L.” How come this paper be proposed as circular economy and zero waste? Please explain more on techno-economical perception.
  4. Please provide possible applications for CNC polymorph II, there an advantage and use in global market.
  5. Comparative table providing a list of references (best 10) providing different methodology to synthesis CNC polymorph II, quantity (how much CNC produced/g of biomass), purity, etc.

Author Response

Reviewer 1:

We appreciate the useful comments of the reviewer regarding the manuscript. Here we are providing the responses to the referees’ comments, the explanation, point by point, and the details of the revisions indicating the corresponding lines in the revised uploaded manuscript.

The manuscript provides good insight toward the production of CNC from Cynara scolymus L and studied the crystallinity using XRD and TEM and provided charesteristic functional groups presence through FTIR. Overall the manuscript can be accepted for publication provided following queries raised below:

1          Regarding the crystallinity index (CI%), as authors suggested CNC-S has better crystallinity compared to CNC-NS, why there is no attempt to improvise the crystallinity of CNC-NS with other known methodology?

Thank you for the interest in our research and the useful comments. Regarding this request, it is useful to observe that once the CNC has been obtained (via Soxhlet or non Soxhlet aparatus) it is not possible to improve the crystallinity Index in a second step, because it depends on how much amorphous part has been eliminated during the procedure. The aim of this research is to explore a new synthetic path, i.e. a rapid and economic process to obtain stable colloid suspension of cellulose nanocrystals from agricultural wastes, without using harmful solvents and Soxhlet apparatus. With our system it is possible to use Cynara wastes to produce cellulose nanocrystals of good quality, even if with a bit lower CI, and acceptable yield, encouraging the recycling and resource reusing of agri-food residues.

2          We would be happy to see Calculations of CNCs yield and residue yield for all four type.

The requested data have been inserted into the experimental part of the manuscript, citing the references to obtain the values. Lines 201-212

3          As author claims, “an approach of circular economy “zero waste” by alternative green extraction of cellulose nanocrystals polymorph II from Cynara scolymus L.” How come this paper be proposed as circular economy and zero waste? Please explain more on techno-economical perception. 

We thank the referee for the useful comment: Agriculture wastes are contributing to the environmental pollution. On the other hand, new materials with interesting properties and applications can be produced starting from biodegradable and renewable biomass. Cellulose nanocrystals CNC are one of these interesting materials. Many industrial companies have developed procedures to produce CNC starting from several materials and CNC are now commercially available in some countries. Nevertheless, the research is still very active in finding new ways to produce this material, as pointed out in many papers and reviews, also starting from lignocellulosic biomass of agri-food origin. The efficient separation of the constituent components of these biomasses is one of the major obstacles to useful use of renewable resources. The objective of this work is to find a rapid and economic process to obtain stable colloid suspension of cellulose nanocrystals from agricultural wastes. In this research we have explored a way to use Cynara wastes to produce cellulose nanocrystals of good quality, encouraging the recycling and resource reusing of agri-food residues, especially in the regions where Cynara cultivation is widespread. This topic has been further evidenced in the introduction of the manuscript adding several references in addition to those already indicated. Lines 36-42 and references therein

4          Please provide possible applications for CNC polymorph II, there an advantage and use in global market

We thank the referee for the useful comment. It is well known that CNC I has better mechanical properties, on the other hand CNC II seems to be more efficient in terms of functionability as well as in better thermal stability which is an essential requirement to be used as additive or filler in composite materials. This arises also from the stronger Hydrogen bond network in CNC-II compared to CNC-I. For this reason CNC-II can be used in the development of membranes for water pollution, which once functionalized can be used as adsorbers for cations and organic dyes. Many papers have been published regarding the desirable application of functionalized CNC in the preparation of biobased innovative materials. This part has been inserted into the introduction of the manuscript,

Lines 93-101

5          Comparative table providing a list of references (best 10) providing different methodology to synthesis CNC polymorph II, quantity (how much CNC produced/g of biomass), purity, etc.

We thank the referee for the useful suggestion. A table containing the requested information has been inserted in the introduction of the manuscript Lines 101-104

Author Response

We appreciate the useful comments of the reviewer regarding the manuscript. Here we are providing the responses to the referees’ comments, the explanation, point by point, and the details of the revisions indicating the corresponding lines in the revised uploaded manuscript.

Manuscript title: Green extraction of cellulose nanocrystals polymorph II from Cynara scolymus L.: challenge for “zero waste” economy Manuscript ID: 1671565 The authors extracted CNCs from an agri--food waste, Cynara scolymus L using two different procedures (i) Soxhlet extraction in ethanol/toluene mixture, and (ii) water boiling of the agricultural waste. Cynara scolymus L. waste was found to produce high quality cellulose nano- crystals as a green alternative to the commonly used synthetic route. The topic is interesting and could promote circular economy (zero-waste approach). However, the novelty of the current study has to be spelt out, indicating what other researchers have done and the gap the authors are aiming to bridge. Therefore, the manuscript should only be accepted for publication after major revision. The following comments and suggestions should be addressed;

1   General editing of the manuscript is required for grammatical errors and language structure

The entire manuscript has been checked out for English language

2   Abstract: The characterization techniques employed in this study should be stated

This part has been inserted in the abstract Lines 18-20

3   What gaps are the authors trying to bridge? No review of what has been done in literature and how this current study differs. The authors should simply state the novelty of this study in relation to the literature

We thank the reviewer for the interest in our research and the useful comments. Regarding this request, it is useful to recall that agriculture wastes are contributing to the environmental pollution. On the other hand, new materials with interesting properties and applications can be produced starting from biodegradable and renewable biomass. Cellulose nanocrystals are one of these interesting materials. Many industrial companies have developed procedures to produce them starting from several materials, and CNC are commercially available in some countries. Nevertheless, the research is still very active in finding new ways to produce this material, as pointed out in many papers and reviews, also starting from lignocellulosic biomass of agri-food origin. The efficient separation of the constituent components of these biomasses is one of the major obstacles to useful use of renewable resources. The objective of this work is to find a rapid and economic process to obtain stable colloid suspension of cellulose nanocrystals from agricultural wastes. In this research we have explored a way to use Cynara wastes to produce cellulose nanocrystals of good quality, encouraging the recycling and resource reusing of agri-food residues, especially in the regions where Cynara cultivation is widespread. This topic has been further evidenced in the introduction of the manuscript adding several references in addition to the ones already indicated. Lines 36-42 and references therein

4   Line 98-99: The characterization techniques used must be defined and authors must indicate the purposes of the techniques (what each of them measures)

The purposes of the techniques used have been added in the text as requested by the reviewer

5   Line 175: “mg/ml” should be written as “mg/mL”.

This has been corrected in the manuscript as requested by the reviewer

6   Line 177-184: The specific purpose for using TEM analysis was not mentioned as well as FTIR, XRD, and TGA. Authors should state the purpose of using these characterization techniques.

The purposes of the techniques used have been added in the text as requested by the reviewer

7   Line 253: The authors should provide the dimension (length) of each of the samples and justify the variations.

We agree that it would be useful for the reader to know the dimensions of the fibers of the samples: unfortunately, in addition to the sensitivity to the beam that prevented to obtain good quality images, we encountered some other difficulties in CNC-S and CNC-NS samples in comparison to the CNC-CF one. The first presented the cellulose fibers always agglomerated, so that it was not possible to distinguish single fibers. The latter presented the fibers embedded in amorphous material. Nevertheless, by an accurate comparison of all the pictures, an average length and thickness of the fibers could be estimated, as reported in the text. We did not find significant variations between the samples.

8   Authors should provide clearer images of samples CNC-CF, CNC-S, and CNC-NS in Figure 2.

We agree with the observation of the reviewer about the not completely satisfying quality of the images, the reason for which has been explained in the manuscript. Due to the lack of contrast of the cellulose in the conventional TEM imaging mode, it had been necessary to use the STEM Z-contrast imaging mode. In this mode a focused beam is used. CNC-S and CNC-NS samples were extremely sensitive to the electron beam damage. Extremely short beam dwell time had to be used to acquire the images, leading to a bad signal to noise ratio. Nevertheless the pictures are readable and representative of each sample.

9   The findings in this current study must be compared with literature under each discussion. Consistencies of the findings with the literature must be emphasized.

The comparison with literature has been inserted in each discussion in addition to the previously indicated references, as requested by the referee (references 2-4, 33-46, 52-53 and 60-61 have been added as requested by the reviewer).

10 Did the authors calculate the crystallinity and yield of the synthesized CNCs? If yes, the results should be provided.

The data about CI are inserted into Table 3. Yields have been provided in the experimental part Lines 201-212

11 The title of the Y-axis for the left image of Figure 5 must be provided

The part has been corrected